# Using step selection functions to analyse human mobility using telemetry data in infectious disease epidemiology: a case study of leptospirosis

Pablo Ruiz Cuenca[1]*, Fábio N Souza[2,3], Roberta Coutinho do Nascimento[3], Ariane Goncalves da Silva[2], Max T Eyre[4], Juliet O Santana[2,3], Daiana de Oliveira[2,3], Emile V Ribeiro de Souza[2], Fabiana G Palma[2], Diogo C de Carvalho Santiago[2], Priscyla dos Santos Ribeiro[5,6], Priscilla Elizabeth Ferreira dos Santos[2], Hussein Khalil[7], Jonathan M Read[1], Cleber Cremonese[2], Federico Costa[2,3,8†], Emanuele Giorgi[1†]

[1]Centre for Health Informatics, Computing, and Statistics (CHICAS), Lancaster Medical School, Lancaster University, Lancaster, United Kingdom; [2]Institute of Collective Health, Federal University of Bahia, Salvador, Brazil; [3]Instituto Gonçalo Moniz, Fundação Oswaldo Cruz, Ministério da Saúde, Salvador, Brazil; [4]Environmental Health Group, London School of Hygiene and Tropical Medicine, London, United Kingdom; [5]Federal University of Bahia, National Institute of Science and Technology in Interdiciplinary and Transdiciplinary Studies in Ecology and Evolution, Salvador, Brazil; [6]Amsterdam University Medical Centre, Leptospirosis Reference Center, Medical Microbiology and Infection Control, Amsterdam, Netherlands; [7]Department of Wildlife, Fish and Environmental Studies, Swedish University of Agricultural Sciences (SLU), Uppsala, Sweden; [8]Department of Epidemiology of Microbial Diseases, School of Public Health, Yale University, New Haven, United States

*For correspondence:
p.ruizcuenca@lancaster.ac.uk

†These authors contributed equally to this work

## eLife Assessment

This study makes a novel and **valuable** contribution by adapting step selection functions, traditionally used in animal ecology, to explore human movement and environmental risk exposure in urban slums, offering a promising framework for spatial epidemiology, particularly regarding leptospirosis. The integration of GPS telemetry with environmental data and the stratification by gender and serostatus are notable strengths that enhance the study's relevance for public health applications. The strength of evidence is **compelling**.

## Abstract

**Background:** Human movement plays a critical role in the transmission of infectious diseases, especially those with environmental drivers like leptospirosis—a zoonotic bacterial infection linked to mud and water contact. Using GPS loggers, we collected detailed telemetry data to understand how fine-scale movements can be analysed in the context of an infectious disease.

**Methods:** We recruited individuals living in urban slums in Salvador, Brazil, to analyse how they interact with environmental risk factors such as domestic rubbish piles, open sewers, and a local stream. We aimed to identify differences in movement patterns inside the study areas by gender, age, and leptospirosis serological status. Step selection functions, a spatio-temporal model used in

animal movement ecology, estimated selection coefficients to represent the likelihood of movement toward specific environmental factors.

**Results:** With 128 participants wearing GPS devices for 24–48 hr, recording locations every 35 s during active daytime hours, we segmented movements into morning, midday, afternoon, and evening. Our results suggested women moved closer to the central stream and farther from open sewers compared to men, while serologically positive individuals avoided open sewers.

**Conclusions:** This study introduces a novel method for analysing human telemetry data in infectious disease research.

**Funding:** Funding provided by Wellcome Trust, UK Medical Research Council, Brazilian National Research Council, Reckitt Global Hygiene Institute, and National Institute of Allergy and Infectious Diseases.

## Introduction

GPS loggers are a growing tool for capturing both human and animal movements (*Owers et al., 2018*; *Fornace et al., 2019*). These small devices can be worn by individuals and record locations at regular preset time intervals. Compared to other methods of collecting human movements, such as cell tower traffic or Google Location History which are suited for analysing large-scale mobility (*Kraemer et al., 2020*; *Moncayo-Unda et al., 2023*), these devices can capture very fine-scale movements. These data are crucial in quantifying exposure within complex environments, where terrain can change rapidly. Furthermore, movements recorded by GPS loggers can be assigned to specific individuals. This allows linkage between individual socio-demographic factors and the data collected, especially convenient when performing epidemiological analyses. Other methods for measuring human mobility are inherently anonymous and do not allow this connection to be made. An important challenge when using GPS loggers is that they rely on individual compliance for carrying the device at all times, an issue which is overcome by the other methods mentioned above.

The analysis of human telemetry data is an emerging field of research in epidemiology. Whilst previous methods have advanced this area of research, improvements could be made. For example, the methods used by *Owers et al., 2018*, to assess the relationship between urban slum residents' movements and the risk of leptospirosis infection were able to analyse differences between genders, but did not consider other important socio-demographic factors. In another study, *Fornace et al., 2019*, used GPS loggers to assess human exposure to mosquito vectors of *Plasmodium knowlesi* malaria and environmental factors associated with this. Various individual-level factors were included in the analyses performed in this paper, questioning how these could affect participants' movements. However, by not including comparisons of possible choices an individual could have made, this study could not determine how the environment may have influenced movement.

Leptospirosis is a zoonotic bacterial infectious disease with strong environmental drivers. It has been estimated to cause over 1 million yearly human cases worldwide, leading to 58,900 deaths (*Costa et al., 2015*). Rats are the main reservoir of the disease, shedding bacteria in their urine (*Adler and de la Peña Moctezuma, 2010*). Human infection is associated with exposure to contaminated waters and soils (*Adler and de la Peña Moctezuma, 2010*; *Johnson et al., 2024*; *Reis et al., 2008*). Evidence shows that in urban slum settings, men have a higher infection risk than women (*Hagan et al., 2016*). This has been attributed to differences in behaviours, especially in how individuals move through their communities, rather than biological differences. Indeed, there is evidence that men tend to visit much larger areas during their daily journeys than women (*Owers et al., 2018*).

Exactly where people are most exposed to high leptospirosis contamination, and therefore where infection is most likely to occur, has not been investigated. Previous studies have focused on the assessment of the peri-domiciliary environment and its associations to infection risk (*Johnson et al., 2024*; *Reis et al., 2008*; *Hagan et al., 2016*). However, these analyses assume people are mostly exposed to infection risk in this area and ignore the exposure that individuals may incur when they move further away from their households. Furthermore, people's movement patterns may differ depending on individual socio-demographic factors which could in turn affect their risk of exposure. If individuals traverse highly contaminated areas where the risk of exposure is heightened, it can lead to an increased risk of infection. This is particularly important in environmentally heterogeneous areas, such as urban slums, where the landscape can change drastically in small spaces. Technological

**eLife digest** Leptospirosis is a disease caused by *Leptospira* bacteria and can be transmitted to humans from other animals. It spreads through the urine of infected animals and can infect individuals who come into contact with contaminated water or soil.

Previous research indicates that in urban slum settings, men face a higher risk of infection than women, which is believed to result from differences in behavior and access to certain locations rather than biological factors. However, data on human movement is typically gathered using mobile data or Google Location History, which often lack detailed information needed to understand movement and behavior at a more refined scale.

GPS loggers are a growing tool for tracking animal movement. These small devices can be worn by individuals and record locations at regular preset time intervals, providing a much more detailed picture than conventional methods. Ruiz Cuenca et al. sought to determine if it was possible to analyze people's movement through their neighborhoods by adapting existing methods used in ecology.

For the movement analysis study, the researchers recruited adults who had been living in one of the study areas in Salvador, Brazil, for at least 6 months. People were tested for potential *Leptospira* infection and were asked to wear GPS loggers for continuous periods of up to 48 hours between March and November 2022. The GPS loggers recorded their location every 35 seconds.

A target of 30 people per study area was chosen, balanced by gender and blind to their infection status. The analysis further focused on three environmental settings: community stream, open sewers and domestic rubbish piles. Ruiz Cuenca et al. used Step Selection Functions (SSFs), a relatively new model for studying the resource selection of animals moving through a landscape. The model compares the environmental attributes of observed steps with alternative random steps taken from the same starting point.

The analyses indicate that step selection functions can be adapted to study how people travel through their neighborhoods. Although the methods used are still novel and results are not conclusive, no apparent difference in movement could be found between infection statuses or ages concerning the distances to stream, open sewer points or domestic rubbish piles. However, women tended to move closer to the central stream and farther from open sewer points than men, suggesting that women may avoid open sewers due to perceived risks, while men may not share these perceptions. Moreover, infected individuals were more likely to move outside the buffer zone for open sewers compared to non-infected individuals.

Leptospirosis is strongly linked to human dwellings, and living near an open sewer may increase the risk of getting infected. A better understanding of how the movement of individuals could affect their risk of infection may enable the implementation of appropriate measures to reduce infection risk. However, further research is needed to fully understand where infections are happening, for example, by increasing the number of people participating in a study and evaluating perceived infection risks.

advances now allow us to record and analyse fine-scale movements to understand how these may affect infection risk.

In this paper, we developed a modelling framework to understand how telemetry data can be used to identify and quantify determinants of human movements, adapting methods from animal movement ecology. We present a novel method for analysing telemetry data to estimate environmental selection as individuals move through their urban communities. This method is applied in a low-income urban setting in Salvador, Brazil, and is used to examine how individuals interact with various key points in their surrounding environment. Furthermore, we analyse if there are any differences in movements inside the study areas between genders, ages, and leptospirosis serological status. This method of analysis overcomes limitations from other studies by, firstly, specifically modelling choice of movement in relation to environmental factors and, secondly, incorporating multiple socio-demographic factors which allows regression relationships to be jointly adjusted for these.

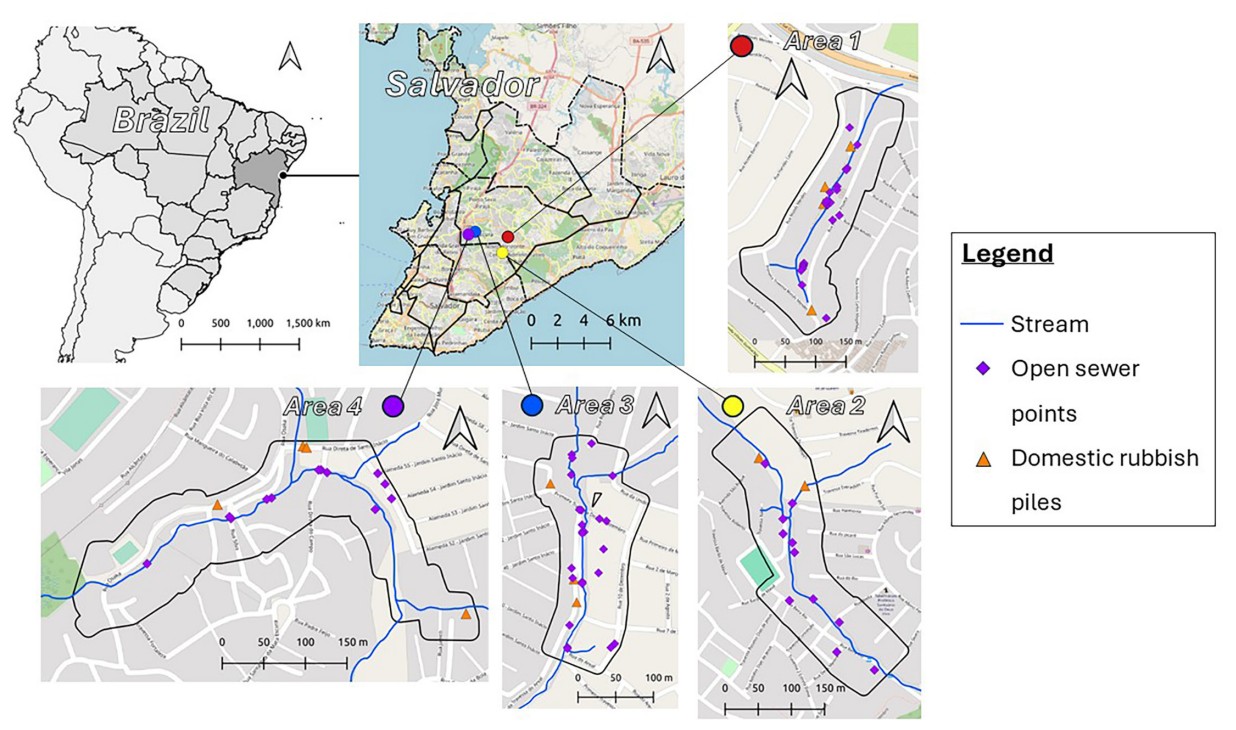

**Figure 1.** Map showing location of each study area in Salvador. Each area includes symbology for stream (blue line), open sewer points (purple diamond), and domestic rubbish piles (orange triangle).

## Methods

ICMJE guidelines have been followed and a STROBE checklist is included with the manuscript as a supplementary material.

### Study areas

This study was nested in a prospective cohort study taking place across Salvador, Brazil (*Cremonese et al., 2023*). Salvador is the third largest city in Brazil, located in the north-eastern region of the country and has a tropical climate. The study areas are considered urban slums (locally called 'favelas'). They were selected for a number of reasons: firstly, they all have similar demographic and socio-economic factors within their populations; secondly, they all have a stream running through the centre of the community, which is considered contaminated; and thirdly, there is a high burden of leptospirosis in these populations.

All four study areas are small, with an approximate size of 0.03 km². They are located across the outskirts of Salvador (*Figure 1*). The communities have very heterogeneous environments, with rapid changes in both land cover and slope. Buildings in these communities have been built with limited or no urban planning. They can be of varying quality, ranging from gated areas with multiple dwellings protected from rain and flooding to single brick buildings with informal entryways.

### Individual characteristics

The eligibility criteria for inclusion in the study were: individuals who (1) had been living at one of the study areas for at least 6 months, (2) slept there at least 3 nights a week, (3) were at least 18 years old, and (4) gave written consent (*Cremonese et al., 2023*). Participants were asked to answer a baseline survey which collected their demographic, social, and economic characteristics, including age and gender. A blood sample was taken from each participant to determine serological evidence for *Leptospira* infection using the microscopic agglutination test (MAT), the standard test used for leptospirosis diagnosis (*Adler and de la Peña Moctezuma, 2010*). In this analysis, a MAT showing antibodies with a titre >1:50 against any *Leptospira* serovar was considered a positive result. Further details about the

laboratory work carried out are available in Appendix 1. The location of their household was recorded and georeferenced by the research team.

Participants who were already enrolled in the cohort study were recruited to take part in the movement analysis study. At the time of recruitment, we found no published scientific studies detailing how to perform sample size calculations for research using GPS data in humans. Therefore, we opted to use convenience sampling instead. A target of 30 people per study area, balanced by gender and blind to their serological status, was chosen for this study.

## GPS data

Individuals who consented to take part in this study were asked to wear GPS loggers for continuous periods of up to 48 hr, which could be repeated. The GPS loggers used were i-got U GT-600, set to record their location every 35 s. We used the manufacturer's software to programme the devices. Data were collected between March and November 2022.

Once the GPS telemetry data was collected, participants' recorded locations were cleaned so as to retain only relocations within the study area boundaries that were recorded between 5 am and 9 pm. This period generally corresponds to an individual's active hours. Interactions with environmental factors outside of the study area boundaries could not be considered in the analysis because high-resolution environmental data outside of the study areas was not available. Individuals with less than 50 relocations within the study area were excluded from the analysis to ensure good model convergence. Details of these excluded individuals can be found in Appendix 1.

## Environmental data

This analysis focused on three environmental factors: community stream, open sewers, and domestic rubbish piles. The latter factor represented areas where rats were more likely to be found, whilst the other factors represented risks of having close contact with *Leptospira* contaminated muds or waters. The location of these different points of interest in the study area was mapped by trained research teams.

These environmental factors were included in analyses in two ways: using distance rasters and buffer rasters. A 1 m resolution raster was created for each environmental factor by calculating the nearest distance for each pixel to the reference points. The buffer rasters, one for each factor, were created using a 20 m buffer around each reference point. The size of this buffer was decided after visiting the study areas and represented an area within which it could be considered a strong interaction with the point of interest. All pixels within this buffer were assigned a value of 1, whilst those outside were given a value of 0. Buffers were used to understand the effect of the immediate vicinity

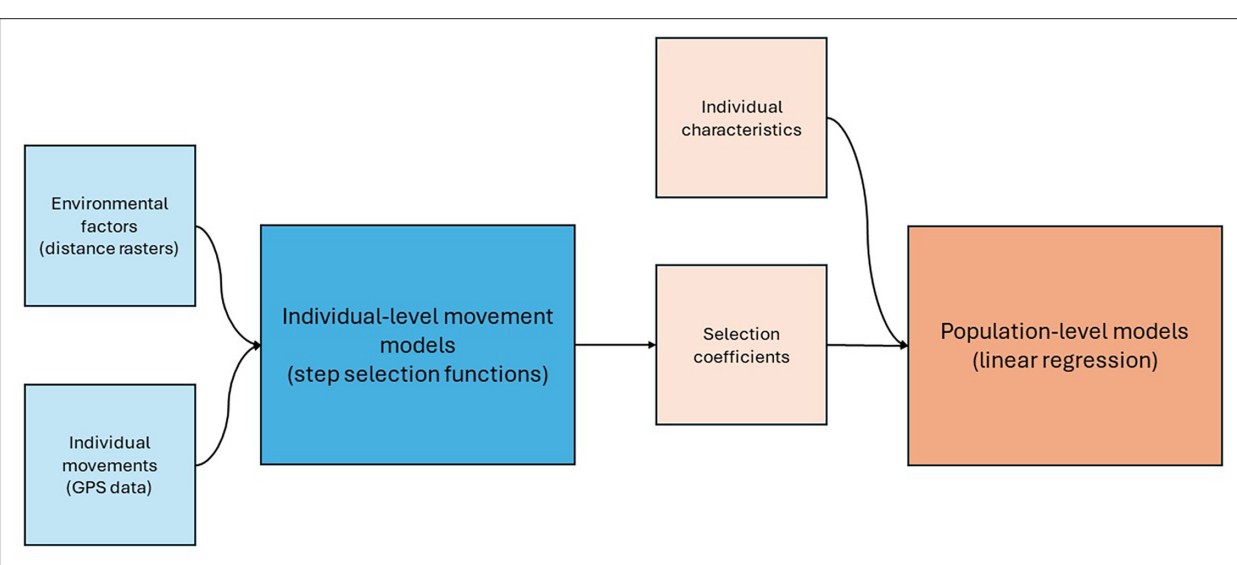

**Figure 2.** Schematic diagram showing what data sources are used in which model, and how models are linked with each other. The blue sections represent phase 1, the individual-level models, whilst the orange section represents phase 2, the population-level model.

of each reference point on movement behaviours. Buffer rasters were also created for each individual's household location, with a 10 m buffer around each location. This represented space within and immediately outside each house. This buffer size accounted for the size of dwellings in these study areas.

## Movement analysis

The analysis was performed in two phases (*Figure 2*). Firstly, each individual's data was analysed alongside the environmental factors. This phase created a set of parameters—called selection coefficients—for each individual. These selection coefficients were specific to each of the environmental factors. In the second phase, the selection coefficient for a particular environmental factor was analysed across the study population. This phase incorporated the individual characteristics for each participant: gender, age, and *Leptospira* serological status. These phases are detailed below. All analyses were carried out in R, version 4.2.1 (*R Development Core Team, 2024*) using tools from tidyverse (*Wickham, 2023*). Specific movement analyses were carried out using package amt (*Signer et al., 2024*).

### Phase 1: Individual-level model

Drawing from the current methodological developments in animal movement ecology, we used step selection functions to characterise individuals' movement behaviours in relation to the environmental factors described above. Step selection functions are a type of movement analysis method that falls under the Resource Selection umbrella. They can also be classified as spatio-temporal point process models (*Fieberg et al., 2021*). In these models, an individual's location at time point $i$ ($\mu_i$) is conditioned on the previous location it was in ($\mu_{i-1}$), the selection coefficients of the environment ($\beta$), and the available space the individual could have travelled to ($\theta$).

$$[\mu_i|\mu_{i-1}, \beta, \theta] \equiv \frac{g\left(x\left(\mu_i\right), \beta\right) f\left(\mu_i|\mu_{i-1}, \Delta_i, \theta\right)}{\int g\left(x\left(\mu\right), \beta\right) f\left(\mu|\mu_{i-1}, \Delta_i, \theta\right) d\mu}$$

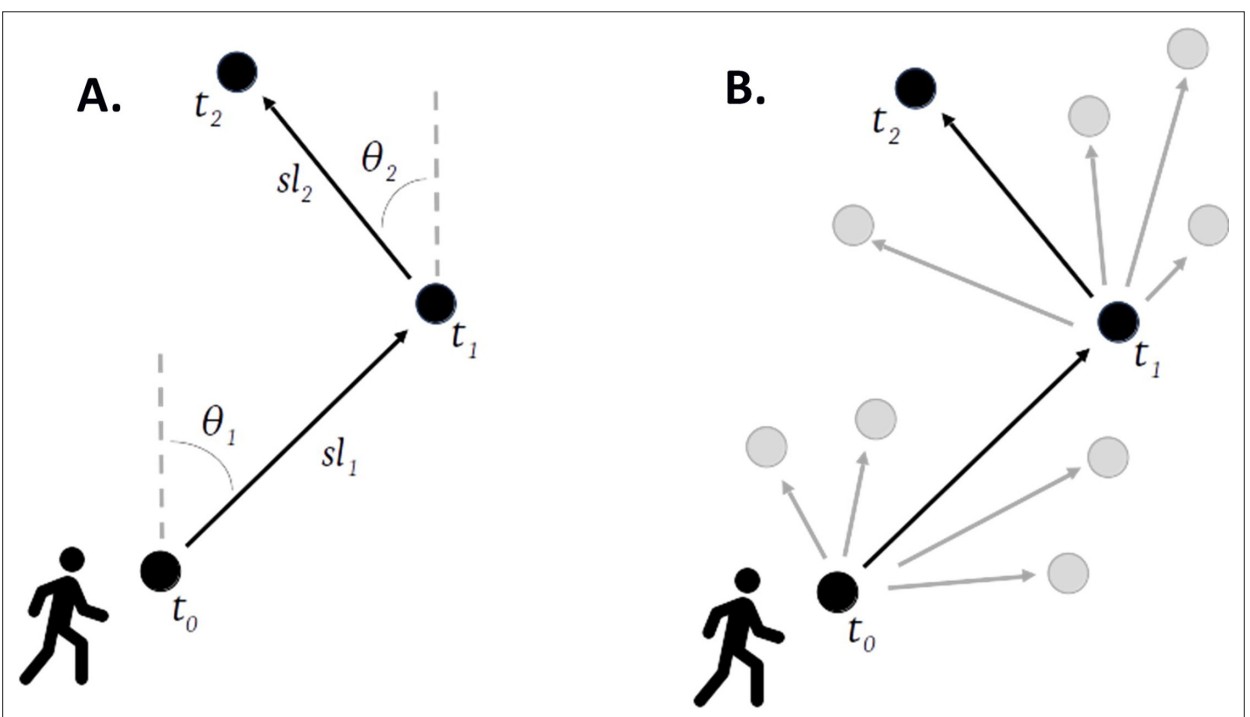

**Figure 3.** Descriptive diagram of step selection functions. (A) Step lengths (*sl*) and turning angles ($\theta$) are used to characterise an individual's movements. (B) These parameters are used to create a set of available steps (grey dots) for every used step (black dots).

Step selection functions have two important components: the availability function ($f(\ldots)$) and the selection function ($g(\ldots)$). The availability function defines the available space that an individual could move inside of within a set of space and time constraints. The selection function specifies how the individual responds to the environmental factors that are close to them when choosing their path, creating a set of selection coefficients for each factor—or resource—included in the model. These selection coefficients are specific to a given individual. This latter component is the focus of our analysis, whilst the former availability function was pre-defined using the empirical data.

The availability function was fitted separately to each recorded location. The step lengths and turning angles between consecutive steps were used to parametrise movement characteristics for an individual (*Figure 3A*). Using these characteristics, a group of available steps (*Figure 3B*, grey dots) was created for each used step (*Figure 3B*, black dots). These represented locations that were consistent with human movements that an individual could have travelled to but chose not to. A total of 100 available steps were created for each used step (*Figure 3B*).

Each individual's telemetry data was analysed by time periods within daytime active hours. These were periods of 4 hr, representing morning (05:00–09:00), midday (09:00–13:00), afternoon (13:00–17:00), and evening (17:00–21:00) activities. Movements across the whole daytime period were also analysed (05:00–21:00). This analysis was performed to examine the effects of circular journeys, when people travel to and back from a same place using a very similar route. By looking at specific time periods, we hoped to capture one-way journeys. As with the full-day analysis, any individuals with less than 50 relocations within the period of analysis were removed from the models.

A conditional logistic regression was used to estimate the selection coefficients for each of the environmental variables for a given individual. A separate model was used for each time period.

$$logit\left(p\right) = \beta_0 + \beta_1 x_1 + \beta_2 x_2 + \beta_3 x_3 + \beta_4 x_4 + \beta_5 x_5 \beta_6 x_6 + \beta_7 x_7 + \beta_8 x_8 + \alpha_{stratum_j}$$

The model estimated the odds of a step being used compared to it being available and unused ($p$), with a logit transformation ($logit\left(p\right)$). The first three variables included in the model ($x_1 - x_3$) represented the different environmental factors (central stream, open sewers, and domestic rubbish piles) and their corresponding selection coefficients ($\beta_1 - \beta_3$). Distance rasters and buffer rasters were included in separate models. The household buffer rasters were included in the next variable ($x_4$). The following three variables ($x_5 - x_7$) represent the movement characteristics of the individual: the step length ($sl$), the natural logarithm of the step lengths ($log\left(sl\right)$), and the cosine of the turning angle ($cos\left(\theta\right)$). These are the same movement characteristics used to create the set of available steps. The final variable included ($x_8$) was the hour within which each step was recorded. The model was stratified by each used step ($\alpha_{stratum_i}$), where $j$ represents each used step and its associated available steps. This model estimates a selection coefficient for each of the environmental factors of interest, conditioned on all other environmental factors, the individual's household location, the individual's movement characteristics, and the hour of the day. These selection coefficients can be interpreted as the likelihood of moving into a specific environmental condition whilst keeping other environmental factors, movement characteristics, and hour of the day constant. For distance rasters, the selection coefficient represents odds of moving further away from the reference point. For buffer rasters, it represents the odds of moving inside of the 20 m buffer of each reference point.

## Phase 2: Population-level model

To assess movement differences between individual characteristics, a population-level linear regression model was used. Separate models were created for each of the three environmental factors, using their corresponding selection coefficients as the outcome, and for each time period (whole daytime period, morning, midday, afternoon, and evening). We used two main groups of models: (1) those assessing differences between genders and ages, which were conditioned on both of these variables and the study area; (2) those assessing differences between *Leptospira* antibody statuses, which were conditioned on gender, age, and study area. The shared equation for each of the models is defined as follows:

$$\hat{\beta}_k = \gamma_0 + \gamma_1 x_1 + \gamma_2 x_2 + \gamma_3 x_3 + \gamma_4 x_4 + Z_k$$

**Table 1.** Summary table comparing parent study participants and movement study participants.

| | Parent study participants | | Movement study participants | |
|---|---|---|---|---|
| | $n$=1086 | %/mean | $n$=128 | %/mean |
| **Study area** | | | | |
| 1—Nova Sussuarana | 297 | 27.3% | 32 | 25.0% |
| 2—Arenoso | 246 | 22.7% | 28 | 21.9% |
| 3—Jardim Santo Inacio | 278 | 25.6% | 35 | 27.3% |
| 4—Calabetao | 265 | 24.4% | 33 | 25.7% |
| **Gender** | | | | |
| Female | 454 | 41.8% | 59 | 46.1% |
| Male | 632 | 58.2% | 69 | 53.9% |
| **Age (mean±SD)** | 32.2 | ±19.7 | 39.4 | ±15.4 |
| *Leptospira* **antibody status** | | | | |
| Positive | 94 | 8.7% | 13 | 10.2% |
| Negative | 992 | 91.3% | 115 | 89.8% |

In these models, the outcome was the estimated selection coefficient ($\beta$) for each environmental factor ($k$). The first two variables, $x_1$ and $x_2$, represented gender (taking values 0 for male and 1 for female) and age, used as a continuous variable. The third variable, $x_3$, represented *Leptospira* antibody status, as a binary variable taking values 1 for a positive test and 0 otherwise. As mentioned previously, a positive result was defined as a positive MAT result for any *Leptospira* serovar. The final variable in the model, $x_4$, represented the study area, included to adjust for any unmeasured differences between study areas. The error term, $Z_k$, captured the residuals from the model, which also accounted for any variation between individuals which was not measured, as well as the sampling error inherent to the estimates of the selection coefficients. To account for variation in the standard errors of the selection coefficients, the variance of $Z_k$ was defined as $w_k/\tau^2$, where $w_k$ is the estimated variance of $\beta_k$ which was used to account for the heterogeneity in the estimate of $\beta_k$.

## Results

### Descriptive statistics

There were a total of 130 individuals who consented to take part in this movement study. Of these, 2 individuals were removed from further analysis due to not having sufficient relocations within the study area boundaries during the 5 am to 9 pm period. They were both male, older than 50 and tested negative for leptospirosis serology. The remaining 128 individuals represented 11.7% of the sample population from the parent study (n=1086). Of the participants in the movement study, 59 (46.0%) were female and their ages ranged from 18 to 83, with a median age of 38 and mean age of 39.5 (SD = 15.5). There were 13 individuals (10.2%) who tested positive for *Leptospira* antibodies. Although these proportions were very similar to those present in the larger sample population from the parent study, the individuals in the movement analysis skewed female and older (*Table 1*).

The majority of individuals spent most of their recorded time during their active daytime hours within their study area boundaries. The percentage of recorded time spent within the study area boundaries ranged from 4% to 100%. The mean percentage was 80%, with a median of 91% and a standard deviation of 25%. Females spent less time within the boundaries than men (females: mean = 76%, SD = 28%; males: mean = 83%, SD = 22%). Individuals who had antibodies against *Leptospira* spent the same time within the study area boundaries as individuals with no antibodies (positive: mean = 83%, SD = 26%; negative: mean = 80%, SD = 25%).

The maximum values for the different environmental distance rasters varied across the four study areas. The maximum distance to open sewers was lowest in study area 3 and highest in study area 2 (1: 199 m; 2: 235 m; 3: 80 m; 4: 208 m). Similarly, the maximum distance to domestic rubbish piles

**Table 2.** Proportion of tracked time (full-day period, 9 am to 5 pm) spent within each buffer. Mean (standard deviation).

|  |  | River buffer | Open sewer buffer | Domestic rubbish buffer |
|---|---|---|---|---|
| Total |  | 0.54 (0.33) | 0.33 (0.31) | 0.07 (0.15) |
| Area | 1—Nova Sussuarana | 0.54 (0.29) | 0.43 (0.29) | 0.15 (0.18) |
|  | 2—Arenoso | 0.65 (0.25) | 0.28 (0.29) | 0.03 (0.13) |
|  | 3—Jardim Santo Inacio | 0.36 (0.35) | 0.41 (0.31) | 0.08 (0.18) |
|  | 4—Calabetao | 0.67 (0.31) | 0.21 (0.28) | 0.01 (0.02) |
| Gender | Female | 0.56 (0.32) | 0.37 (0.32) | 0.07 (0.15) |
|  | Male | 0.53 (0.34) | 0.30 (0.29) | 0.06 (0.16) |
|  | Negative | 0.55 (0.33) | 0.35 (0.31) | 0.07 (0.16) |
| Leptospirosis serological status | Positive | 0.48 (0.29) | 0.25 (0.30) | 0.02 (0.05) |

was lowest in study area 3 and highest in study area 2 (1: 214 m; 2: 363 m; 3: 153 m; 4: 247 m). These differences are attributed to the number of open sewer points and domestic rubbish piles within each study area. The maximum distance to the central stream was highest in study area 1 and lowest in study area 3 (1: 217 m; 2: 209 m; 3: 94 m; 4: 172 m).

Similarly, the proportion of tracked time spent within the various environmental buffers varied across characteristics. These can be found in *Table 2*. There were significant differences between characteristics and within characteristics, represented by the high standard deviations. More detailed descriptive statistics are available in Appendix 1.

## Movement analysis

The results from the movement analysis are presented in the odds scale. A positive value represents higher odds of moving towards an increasing value for each raster. As described previously, for distance rasters, this is interpreted as moving further away from the point of reference (*Table 3*), whilst for buffer rasters this is interpreted as moving into the 20 m buffer area for each point of reference (*Table 4*).

We found no differences in how individuals moved with regard to the distance to the central stream by age (OR: 1.00; 95% CI: 1.00, 1.00; p=0.697) or *Leptospira* antibody status (OR: 0.99; 95% CI: 0.96,

**Table 3.** Estimated differences ($\gamma$) in selection coefficients ($\beta$) for each environmental factor using distance-based rasters. Values >1 represent increasing distance from points of reference.

|  | Community stream | | Open sewers | | Domestic rubbish piles | |
|---|---|---|---|---|---|---|
|  | Estimate | 95% CI | Estimate | 95% CI | Estimate | 95% CI |
| **Gender*** |  |  |  |  |  |  |
| Male | (Ref) | – | (Ref) | – | (Ref) | – |
| Female | 0.98 | 0.97, 0.99 | 1.04 | 1.02, 1.06 | 0.99 | 0.98, 1.01 |
| **Age[†]** |  |  |  |  |  |  |
| Per year increase | 1.00 | 1.00, 1.00 | 1.00 | 1.00, 1.00 | 1.00 | 1.00, 1.00 |
| ***Leptospira* serological status [‡]** |  |  |  |  |  |  |
| Negative | (Ref) | – | (Ref) | – | (Ref) | – |
| Positive | 0.99 | 0.96, 1.01 | 1.03 | 1.00, 1.07 | 1.00 | 0.98, 1.02 |

(Ref) is the reference group for the Odds Ratio.

*Adjusted for age and study area.

[†]Adjusted for gender and study area. Values represent increases by 1 year of age.

[‡]Adjusted for gender, age, and study area.

**Table 4.** Estimated differences ($\gamma$) in selection coefficients ($\beta$) for each environmental factor using 20 m buffers around each point of reference.

Values>1 represent movement within the buffer zone for each point of reference.

| | Community stream | | Open sewers | | Domestic rubbish piles | |
|---|---|---|---|---|---|---|
| | Estimate | 95% CI | Estimate | 95% CI | Estimate | 95% CI |
| **Gender*** | | | | | | |
| Male | (Ref) | – | (Ref) | – | (Ref) | – |
| Female | 1.22 | 1.02, 1.46 | 0.95 | 0.80, 1.14 | 0.92 | 0.66, 1.27 |
| **Age†** | | | | | | |
| Per year increase | 1.00 | 1.00, 1.00 | 0.99 | 0.98, 1.00 | 1.00 | 0.99, 1.01 |
| ***Leptospira* serological status ‡** | | | | | | |
| Negative | (Ref) | – | (Ref) | – | (Ref) | – |
| Positive | 0.89 | 0.67, 1.19 | 0.64 | 0.47, 0.87 | 0.85 | 0.48, 1.49 |

(Ref) is the reference group for the Odds Ratio.

*Adjusted for age and study area.

†Adjusted for gender and study area. Values represent increases by 1 year of age.

‡Adjusted for gender, age, and study area.

1.01; p=0.273). Similarly, movements relative to the 20 m buffer for the central stream were the same across ages (OR: 1.00; 95% CI: 1.00, 1.01; p=0.280) and across *Leptospira* serological status (OR: 0.89; 95% CI: 0.67, 1.19; p=0.433). There was evidence that women moved closer to the stream than men, even after accounting for the effects of age, study area, and the location of their households (OR: 0.98; 95% CI: 0.97, 0.99; p=0.003). This effect was more pronounced in the analysis of the 20 m buffered area (OR: 1.22, 95% CI: 1.02, 1.46; p=0.026).

As with the above, there was no evidence of different movement behaviours relative to distance to open sewers by age (OR: 1.00; 95% CI: 1.00, 1.00; p=0.572) or *Leptospira* antibody status (OR: 1.03; 95% CI: 1.00, 1.07; p=0.054). Women were found to move further away from open sewers compared to men (OR: 1.04; 95% CI: 1.02, 1.06; p<0.001). When analysing movements relative to the 20 m buffer around open sewers, we found no evidence of differences between genders (OR: 0.95; 0.80, 1.14; p=0.580). We found evidence of a small tendency to move outside of the 20 m buffer around open sewers as people aged, although the effect could be considered negligible (OR: 0.99; 95% CI: 0.98, 1.00; p=0.003). We also found evidence of a strong inclination for people with *Leptospira* antibodies to move outside of the buffers around open sewers, compared to people with no antibodies (OR: 0.64; 95% CI: 0.47, 0.87; p=0.005).

Our analysis showed no evidence of different movement behaviours relative to the distance to rubbish piles across genders (OR: 0.99; 95% CI: 0.98, 1.01; p=0.280), ages (OR: 1.00; 95% CI: 1.00, 1.00; p=0.466), or *Leptospira* antibody statuses (OR: 1.00; 95% CI: 0.98, 1.02; p=0.760). We also found no evidence when analysing movements relative to the 20 m buffer around rubbish piles across genders (OR: 0.92; 95% CI: 0.66, 1.27; p=0.600), ages (OR: 1.00; 95% CI: 0.99, 1.01; p=0.989), or *Leptospira* antibody statuses (OR: 0.80; 95% CI: 0.44, 1.49; p=0.482).

It is important to highlight that the effect sizes of the selection coefficients for the distance-based rasters (*Table 3*) are very small and could be considered negligible. This may be linked to the spatial scale used, as these values represent increases of 1 m. A coarser scale may have produced larger effect sizes that may have been easier to conceptualise. However, given the focus on fine-scale movement, we decided to keep this spatial scale for the analysis.

## Analysis by time periods

Movements were subdivided into four time periods: morning (5 am to 9 am), midday (9 am to 1 pm), afternoon (1 pm to 5 pm), and evening (5 pm to 9 pm). The demographic characteristics of all individuals removed from analyses for having less than 50 relocations within a specific time period can be

found in Appendix 1. The interactions with the environmental factors were similar to those reported for whole-day activities, although there were some key differences (*Figure 4*).

We found no differences in movements relative to the central stream as people aged or between *Leptospira* antibody status across the four periods. Women still moved closer to the central stream than men across all periods. We also saw that women had a higher tendency to move within the 20 m buffer for the stream compared to men across all periods.

Movement in relation to distance to open sewer points and their respective 20 m buffers showed no difference across all four periods. The strength of the selection effect seen in serologically positive individuals for moving outside of the 20 m buffer varied, with stronger effects seen in the morning and evening periods.

Domestic rubbish piles did not appear to have an effect on movement differences between ages or *Leptospira* antibody status across all periods. We found women moved outside of the 20 m buffer zone more than men during the morning period only. Otherwise, no notable differences were seen.

## Discussion

Our study aimed to apply a novel methodology to the area of human mobility analysis in infectious disease epidemiology, focusing on leptospirosis in four urban slums in Salvador, Brazil. We assessed movements in relation to central streams, open sewer points, and domestic rubbish piles and observed changes throughout the day using step selection functions. These are a modelling approach which we have taken and adapted from animal movement ecology. Our findings showed that step selection functions could be an effective method to identify movement behaviours. To understand how the results could be described in the context of infectious disease epidemiology, we have explained our interpretation of the findings, including strengths and limitations. However, it is important to highlight that, given this is a novel methodology, the evidence we present is not conclusive and further research is required.

The results suggested no movement differences between *Leptospira* antibody statuses or ages concerning the distances to stream, open sewer points, or domestic rubbish piles. Our findings consistently showed that women tended to move closer to the central stream and farther from open sewer points than men, adjusted for age and study area. We also found that women had a tendency to move within the 20 m buffer of the central streams compared to men, and that seropositive individuals were more inclined to move outside of the buffer zone for open sewers compared to seronegative individuals. Movement patterns did not vary significantly throughout the day. Previous research indicates that men in similar communities perceive themselves as less vulnerable to leptospirosis compared to women (*Khalil et al., 2021*). Additionally, a knowledge, attitudes, and practices analysis showed that men have lower scores for both knowledge and attitudes towards leptospirosis and its associated risks (*Palma et al., 2022*). Our findings align with these studies, suggesting that women may avoid open sewers due to perceived risks, while men may not share these perceptions. Social areas, which may have gender differences, also contribute to different movement behaviours. One might conclude that the stream is used for gendered chores such as washing clothes. However, this is not the case in our study areas. Following discussion with residents, we know that they perceive the stream as highly contaminated and avoid using its waters for cleaning or other household chores.

Our results contrast those reported by *Owers et al., 2018*, who found no differences in space use between genders after using GPS loggers to analyse individuals' movements. This discrepancy could be explained by the differences in length of time being analysed. Owers et al. were only able to analyse data collected over 24 hr periods, whereas our analysis was longer and included data collected over periods of up to 48 hr, which could be repeated. The contrasting results could also be attributed to the different populations studied. Although overall these populations resided in very similar communities in Salvador, they could have different characteristics that affect movement behaviours.

Our findings regarding the interactions with rubbish piles may be explained by various reasons. There is evidence that proximity to rubbish piles does not drive *Leptospira* seropositivity in similar areas to those used in our analysis (*Khalil et al., 2021*). Whilst this proximity does increase rat sightings, this reduced effect on infection risk could lead individuals to disregard the locations of rubbish piles when choosing their travel paths. Another possible explanation is that there may be an unmeasured environmental variable that is interacting with the distance to rubbish piles, which needs further

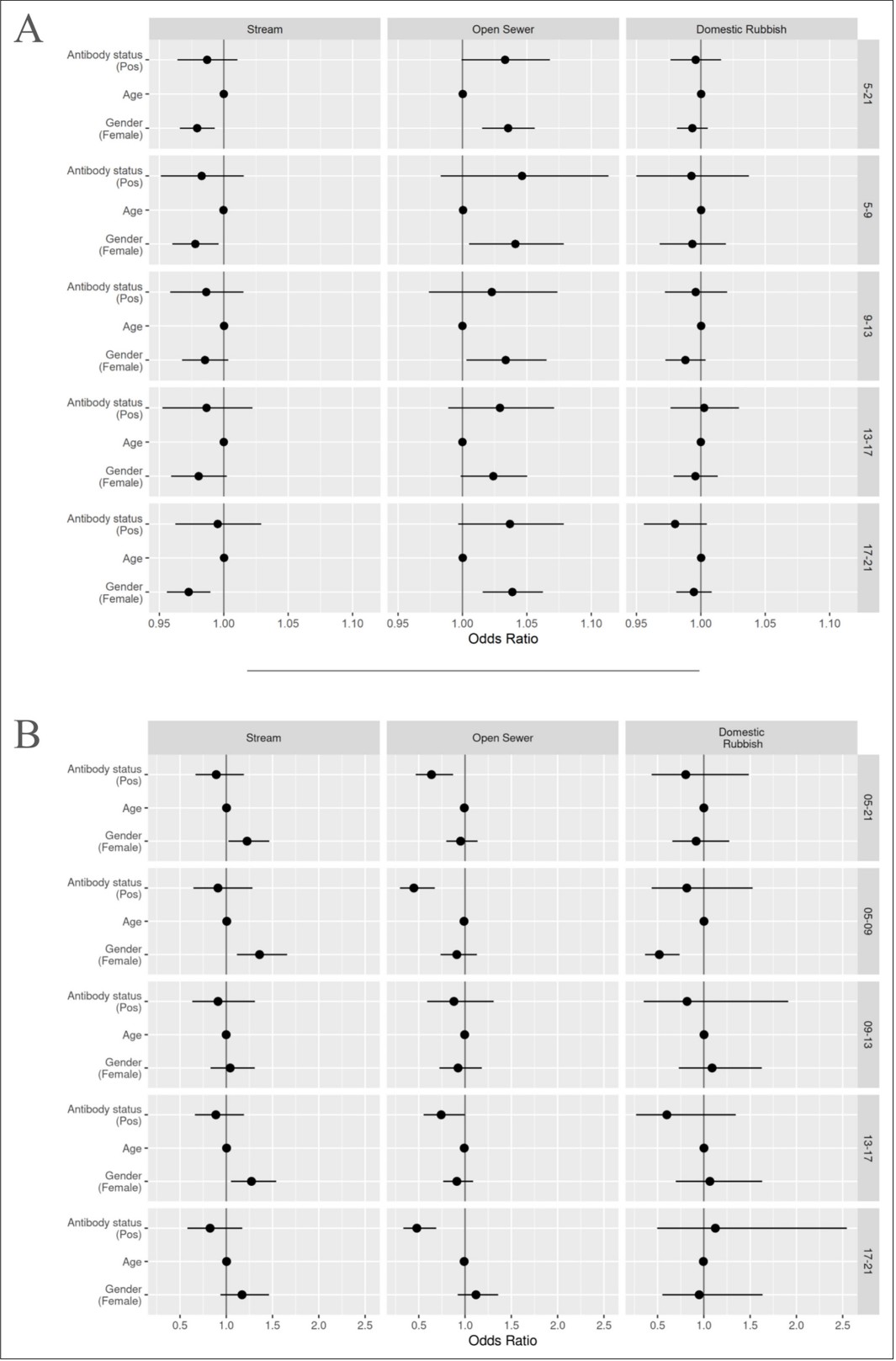

**Figure 4.** Graph showing results of final analyses. (**A**) Results for distance-based rasters; values above 1 interpreted as increasing distance to points of reference. (**B**) Results for 20 m buffer-based rasters; values above 1 show movement within buffer zones. Each horizontal band represents a specific time period (right hand side y-axis label): all day (5 am to 9 pm, *Tables 2 and 3*), morning (5 am to 9 am), midday (9 am to 1 pm), afternoon (1 pm to 5 pm),

*Figure 4 continued on next page*

*Figure 4 continued*
and evening (5 pm to 9 pm). All data points include their corresponding 95% confidence intervals, some of which are too narrow to show up clearly.

investigation. For example, violence could be interacting with where rubbish accumulates. We discuss violence further in a paragraph below.

The evidence showing *Leptospira*-positive individuals avoiding open sewers was surprising. Although we were expecting to see an effect in the opposite direction, showing individuals with *Leptospira* antibodies interacting closely with open sewers, there are a few possible explanations for our findings. If individuals with antibodies are also actively infected, they could be symptomatic and therefore alter their behaviour to avoid high-risk areas. Alternatively, individuals with antibodies could be more aware of risks due to previous infections and display more protective behaviours than people who have not had any previous infections.

During informal conversations with community residents, it became clear that violence plays a key role in individuals' decisions on where they go. Violence in these communities is perceived as hyper-local, restricted to one corner or small square within the communities. It is unclear what drives this perception, but nevertheless, it is an important factor that could be accounted for. Further research is required to develop methods that can capture these perceptions in spatial formats that could be incorporated into similar movement studies. Age did not affect movement choices, suggesting consistent perceptions of environmental risks or stable use of urban spaces across ages.

We expected different movement patterns at various times of day, anticipating circular journeys (an individual going somewhere and back again on the same route). However, our results showed consistent movement patterns, possibly due to the analysis period's length or other unmeasured factors modulating movements. Our results could also be indicative of evidence that strictly circular journeys through these communities, where individuals are travelling through the exact same path for both journeys, are not common, and that movement interactions with urban surroundings do not vary throughout the day.

To our knowledge, this is the first study that uses step selection functions to model movement behaviours in the context of human infectious disease epidemiology. This method has provided quantitative evidence that there may be differences in how men and women move through their communities, strengthening the argument that the variation in leptospirosis exposure and infection risk between genders is due to behavioural differences rather than physiological differences. Additionally, we show that individuals consider environmental features differently when moving through their communities. Highlighting the effects of these variables on movement would not have been possible with the approaches previously used to model human movement. Our approach provides a better understanding of how individuals relate to their surrounding urban environment and how they interact with features that could increase the risk of leptospirosis.

Several important limitations must be highlighted. This study involves a relatively small sample of a larger population, slightly skewed towards older women compared to the parent study. There are few individuals testing positive for *Leptospira* antibodies. As a result, the findings are biased towards the more represented individuals, limiting their generalisability. Additionally, all participants are from specific areas in Salvador, which may further limit the generalisability to similar contexts. Further research is needed to develop appropriate study designs using these methods, including how many individuals should be recruited. The small number of *Leptospira*-positive individuals also makes the estimation for the effect of this characteristic more difficult. We would also like to restate that a positive antibody response to any *Leptospira* serovar does not indicate active infection. A positive result merely indicates that the individual has been infected at some stage, either symptomatically or asymptomatically, and has produced an immune response. Information on the timing of the infection could instead be a variable showing a stronger association with movement. Another important limitation is that we did not collect data on behaviours. If risky or protective behaviours, such as the use of closed footwear, had been available at the appropriate temporal resolution (e.g. hourly intervals), these could have been included in the step selection functions and could have shown significant associations. Although these are important limitations which require cautious interpretation of results, they do not detract from the value of exploring this novel methodology in this context. This methodology

also provides a crucial starting point for exploring how movement characteristics can differ between individuals in these environments.

Step selection functions also have limitations that must be considered. While these methods can model the choice of moving in a specific direction, they do not account for the initial distance from the individual. For instance, an individual moving towards the central stream from far away will have a high selection coefficient for this environmental factor, which does not indicate their starting distance. This is important because environmental risk factors cease to provide risk beyond a certain distance. This limitation was overcome by using buffer zones around specific points of interest, but it is crucial to highlight the importance of correctly interpreting all results. Similarly, step selection functions do not quantify how long an individual spent within this high-risk distance. Additionally, these models have some underlying assumptions that may be violated in this study. Step selection functions assume each step is independent, conditioned on the previous step. This can be violated by circular journeys. Although we attempted to account for these by analysing specific periods of the day, a higher temporal resolution of analysis may be needed if circular journeys are still present within each period. Another assumption is that movement is smooth through the environment. In urban environments, this may not hold true, as street layouts may force sharp corners in movements. The effect of violating this assumption is not immediately clear and requires further methodological research to understand its significance. Finally, we assumed that by including movement characteristics (step lengths and turning angles) into our models, we were accounting for goal-oriented behaviour. These assumptions need to be considered in future studies that attempt to use step selection functions to analyse human mobility.

Despite these limitations, this study has several valuable strengths. By including steps an individual could have taken but did not (i.e. available steps, grey dots in *Figure 3B*), the models allow us to estimate choice. Additionally, the models use each individual's movement characteristics to create these available steps, resulting in a realistic representation of movement behaviours. This creates more realistic estimates of environmental interactions than those created using existing methods.

Another significant strength is the specificity of the individual-level and population-level models. First, the population-level linear regression models allow multiple individual characteristics to be included, producing results that can be adjusted as needed. Although not considered in this study, these models also provide flexibility in the type of variable interactions that can be specified, allowing for non-linear effects if necessary. Second, the individual-level conditional logistic regression models are conditioned for all included variables. This enables the estimation of the selection coefficients for each environmental factor after adjusting for potential confounders. This is particularly useful in our case, as open sewer points are often close to the central stream in all study areas (*Figure 1*).

Overall, we believe this method is a useful tool in analysing human mobility in the context of infectious disease epidemiology. This modelling approach could also be used in other areas of research which analyse human movements and choice relating to surrounding environmental features, such as urban planning. A major benefit of step selection functions is the use of rasters, which provide flexibility when investigating environmental features. Creative uses of rasters could provide interesting questions and results. Although the focus of these models is looking at choice in space, the methods could also be adapted to analyse choice in time (e.g. are there temporal variables that affect when a rat enters a household).

To conclude, we provide a worked example of how to use step selection functions to analyse human movements in the field of infectious disease epidemiology. This highlights the usefulness of adapting methods from other fields to answer questions that would otherwise be difficult to answer with the existing methodology. By doing so, we develop a better understanding of environmental interactions and how to leverage the large datasets provided by GPS loggers. Although our focus was leptospirosis, these methods can be adapted to model the exposure to any disease where movement and the environment play an important role.

## Acknowledgements

We would like to thank all residents from the study areas, without whom this work would not have been able to be completed. The authors would also like to specially thank the GIC (Grupo Impulsor Comunitario) for their support and warmth. This work was supported by the Wellcome Trust, NIHR/ Wellcome Global Health Partnership (218987/Z/19/Z) to FC. FNS received a research scholarship from

the Brazilian National Research Council (CNPq:150142/2024-2). PRC is in receipt of a studentship from the Medical Research Council, UK. ME was supported through a Reckitt Global Hygiene Institute fellowship. JMR acknowledges support from the National Institute of Allergy and Infectious Diseases (Grant number 1R01AI160780-01).

## Additional information

### Funding

| Funder | Grant reference number | Author |
|---|---|---|
| Wellcome Trust | 10.35802/218987 | Federico Costa |
| Conselho Nacional de Desenvolvimento Científico e Tecnológico | 150142/2024-2 | Fábio N Souza |
| Medical Research Council | PhD studentship | Pablo Ruiz Cuenca |
| Reckitt Global Hygiene Institute | Fellowship | Max T Eyre |
| National Institute of Allergy and Infectious Diseases | 1R01AI160780-01 | Jonathan M Read |

The funders had no role in study design, data collection and interpretation, or the decision to submit the work for publication. For the purpose of Open Access, the authors have applied a CC BY public copyright license to any Author Accepted Manuscript version arising from this submission.

### Author contributions

Pablo Ruiz Cuenca, Conceptualization, Data curation, Software, Formal analysis, Investigation, Visualization, Methodology, Writing – original draft, Writing – review and editing; Fábio N Souza, Conceptualization, Resources, Data curation, Software, Formal analysis, Investigation, Writing – original draft, Project administration, Writing – review and editing; Roberta Coutinho do Nascimento, Data curation, Investigation, Writing – review and editing, Field team, laboratory; Ariane Goncalves da Silva, Investigation, Writing – review and editing, Field team, laboratory; Max T Eyre, Formal analysis, Supervision, Writing – review and editing; Juliet O Santana, Data curation, Software, Investigation, Writing – review and editing; Daiana de Oliveira, Data curation, Investigation, Writing – review and editing, Laboratory; Emile V Ribeiro de Souza, Data curation, Investigation, Writing – review and editing, Field team; Fabiana G Palma, Data curation, Validation, Investigation, Writing – review and editing, Field team; Diogo C de Carvalho Santiago, Data curation, Investigation, Writing – review and editing, Field team; Priscyla dos Santos Ribeiro, Data curation, Investigation, Writing – review and editing, Laboratory; Priscilla Elizabeth Ferreira dos Santos, Data curation, Investigation, Writing – review and editing, Field team, laboratory; Hussein Khalil, Validation, Investigation, Writing – review and editing; Jonathan M Read, Supervision, Investigation, Writing – review and editing; Cleber Cremonese, Resources, Data curation, Supervision, Validation, Investigation, Writing – review and editing; Federico Costa, Conceptualization, Resources, Formal analysis, Supervision, Funding acquisition, Validation, Investigation, Project administration, Writing – review and editing; Emanuele Giorgi, Conceptualization, Software, Formal analysis, Supervision, Investigation, Methodology, Writing – review and editing

### Author ORCIDs

Pablo Ruiz Cuenca https://orcid.org/0000-0002-2180-9509
Fábio N Souza https://orcid.org/0000-0002-3542-8918
Max T Eyre http://orcid.org/0000-0001-9847-8632
Juliet O Santana http://orcid.org/0000-0002-1034-6991
Fabiana G Palma http://orcid.org/0000-0002-7345-4914
Cleber Cremonese http://orcid.org/0000-0003-2700-7416
Emanuele Giorgi https://orcid.org/0000-0003-0640-181X

### Ethics
Ethical approval for this study was obtained from the ethics committee at the Collective Health Institute, Federal University of Bahia (CEP/ISC/UFBA) under number CAEE 32361820.7.0000.5030, and the national research ethics committee (CONEP) linked to the Brazilian Ministry of Health under approval number 4.235.251. All participants involved in the study provided written informed consent before data collection.

Reviewer #1 (Public review): https://doi.org/10.7554/eLife.107153.3.sa1
Reviewer #2 (Public review): https://doi.org/10.7554/eLife.107153.3.sa2
Author response https://doi.org/10.7554/eLife.107153.3.sa3

## Additional files

### Supplementary files
MDAR checklist

Source data 1. Data used in population-level models, showing step selection coefficients for each individual and each environmental factor.

Source code 1. R code used for individual-level step selection functions.

Source code 2. R code used for population-level models.

### Data availability
All data are managed in accordance with the ethical standards and data management policies of the affiliated institutions. Due to the inclusion of personal information from survey participants, the datasets used and/or analysed during the current study cannot be publicly shared. Interested researchers can contact Dr Federico Costa (fcosta2001@gmail.com) to obtain the raw data, clearly stating the purpose of their study including the protocol and analysis plan. They will also have to obtain the necessary ethical approvals from the ethics committee at the Collective Health Institute of the Federal University of Bahia and the National Ethics Committee linked to the Brazilian Ministry of Health. Relevant details are available in the ethics section of the manuscript. Anonymized data used for the population-level models has been made available in the supplementary material along with the code used to create it and analyse it. This data represents the selection coefficients of each of the individual-level step selection functions used.

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

# Appendix 1

## Descriptive statistics

### Telemetry data

The mean number of hours of telemetry data provided by an individual was 13.3 hr, with a standard deviation of 13.5 hr. The mean number of locations recorded by the GPS loggers was 2767 points (SD = 1947.2). There were no differences in the number of hours or number of locations recorded by gender, age, or leptospirosis antibody status. There were notable differences in the number of hours recorded and the number of locations by study area. Study area 1 (NVS) had the lowest number of hours recorded (mean = 5.6 hr, SD = 5.6), whilst all other areas had similar hours recorded (area 2: mean = 15.0, SD = 11.4; area 3: mean = 10.9, SD = 14.0; area 4: mean = 20.7, SD = 15.3). The mean number of locations recorded was all similar across all study areas (area 1: mean = 2048, SD = 1206; area 2: mean = 2831, SD = 1302; area 3: mean = 2992, SD = 2737; area 4: mean = 3107, SD = 1761).

### Excluded individuals

**Appendix 1—table 1.** Demographic details of excluded individuals due to having less than 50 relocations.

| ID (anonymised) | Relocation below 50 | Period | Gender | Age group | Leptospirosis serological status |
|---|---|---|---|---|---|
| 60 | TRUE | 05–21 | Male | 50–54 | Neg |
| 91 | TRUE | 05–21 | Male | >55 | Neg |
| 15 | TRUE | 05–09 | Female | 45–49 | Neg |
| 60 | TRUE | 05–09 | Male | 50–54 | Neg |
| 81 | TRUE | 05–09 | Female | 50–54 | Neg |
| 91 | TRUE | 05–09 | Male | >55 | Neg |
| 108 | TRUE | 05–09 | Female | 50–54 | Neg |
| 109 | TRUE | 05–09 | Male | 20–24 | Neg |
| 128 | TRUE | 05–09 | Male | 25–29 | Neg |
| 129 | TRUE | 05–09 | Male | 40–44 | Pos |
| 15 | TRUE | 09–13 | Female | 45–49 | Neg |
| 24 | TRUE | 09–13 | Female | 50–54 | Neg |
| 60 | TRUE | 09–13 | Male | 50–54 | Neg |
| 70 | TRUE | 09–13 | Male | 35–39 | Neg |
| 71 | TRUE | 09–13 | Female | 35–39 | Neg |
| 76 | TRUE | 09–13 | Male | 35–39 | Neg |
| 91 | TRUE | 09–13 | Male | >55 | Neg |
| 108 | TRUE | 09–13 | Female | 50–54 | Neg |
| 109 | TRUE | 09–13 | Male | 20–24 | Neg |
| 128 | TRUE | 09–13 | Male | 25–29 | Neg |
| 129 | TRUE | 09–13 | Male | 40–44 | Pos |
| 24 | TRUE | 13–17 | Female | 50–54 | Neg |
| 60 | TRUE | 13–17 | Male | 50–54 | Neg |
| 71 | TRUE | 13–17 | Female | 35–39 | Neg |
| 76 | TRUE | 13–17 | Male | 35–39 | Neg |
| 91 | TRUE | 13–17 | Male | >55 | Neg |

| ID (anonymised) | Relocation below 50 | Period | Gender | Age group | Leptospirosis serological status |
|---|---|---|---|---|---|
| 5 | TRUE | 17–21 | Male | 35–39 | Neg |
| 7 | TRUE | 17–21 | Male | 50–54 | Neg |
| 18 | TRUE | 17–21 | Male | 30–34 | Neg |
| 22 | TRUE | 17–21 | Male | 45–49 | Neg |
| 24 | TRUE | 17–21 | Female | 50–54 | Neg |
| 27 | TRUE | 17–21 | Male | 30–34 | Neg |
| 30 | TRUE | 17–21 | Female | >55 | Neg |
| 60 | TRUE | 17–21 | Male | 50–54 | Neg |
| 71 | TRUE | 17–21 | Female | 35–39 | Neg |
| 91 | TRUE | 17–21 | Male | >55 | Neg |
| 114 | TRUE | 17–21 | Female | 40–44 | Neg |

## Serological data

Serologically positive individuals were equally distributed across ages and genders, although the oldest male included in the analysis was also serologically positive (*Appendix 1—figure 2*).

There were also no significant skews in the household characteristics relative to the environmental factors being analysed (*Appendix 1—figure 3*).

## Laboratory work

All samples were tested using the MAT test, the reference test for serological diagnosis of leptospirosis, as designated by the WHO. The diagnostic panel used included the following serovars:

- *Leptospira kirschneri* serovar Cynopteri strain 3522C
- *L. kirschneri* serovar Grippothyphosa strain Duyster
- *Leptospira interrogans* serovar Canicola strain H. Ultrech
- *L. interrogans* serovar Autumnlais strain Akiyami A
- *Leptospira borgspetersenii* serovar Ballum strain MUS 127
- *L. interrogans* serovar Copenhageni strain Fiocruz L1-130 (locally isolated in 1996)
- *L. interrogans* serovar Copenhageni strain Fiocruz LV3954

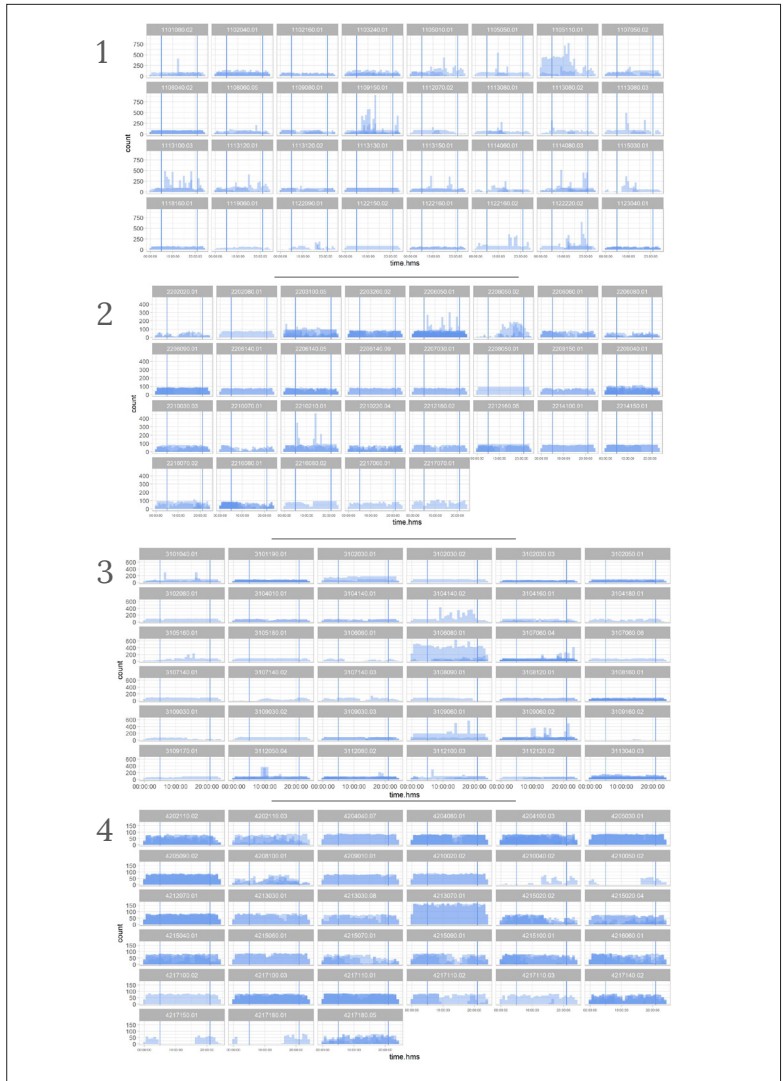

**Appendix 1—figure 1.** Distribution of telemetry data provided by each individual across 24 hr periods (x-axis), separated into each of the four study areas (1: NVS, 2: ARE, 3: JSI, 4: CAL). Overlapping areas represent multiple days. Vertical bars represent 5 am (left-hand bar) and 9 pm (right-hand bar), the period of analysis.

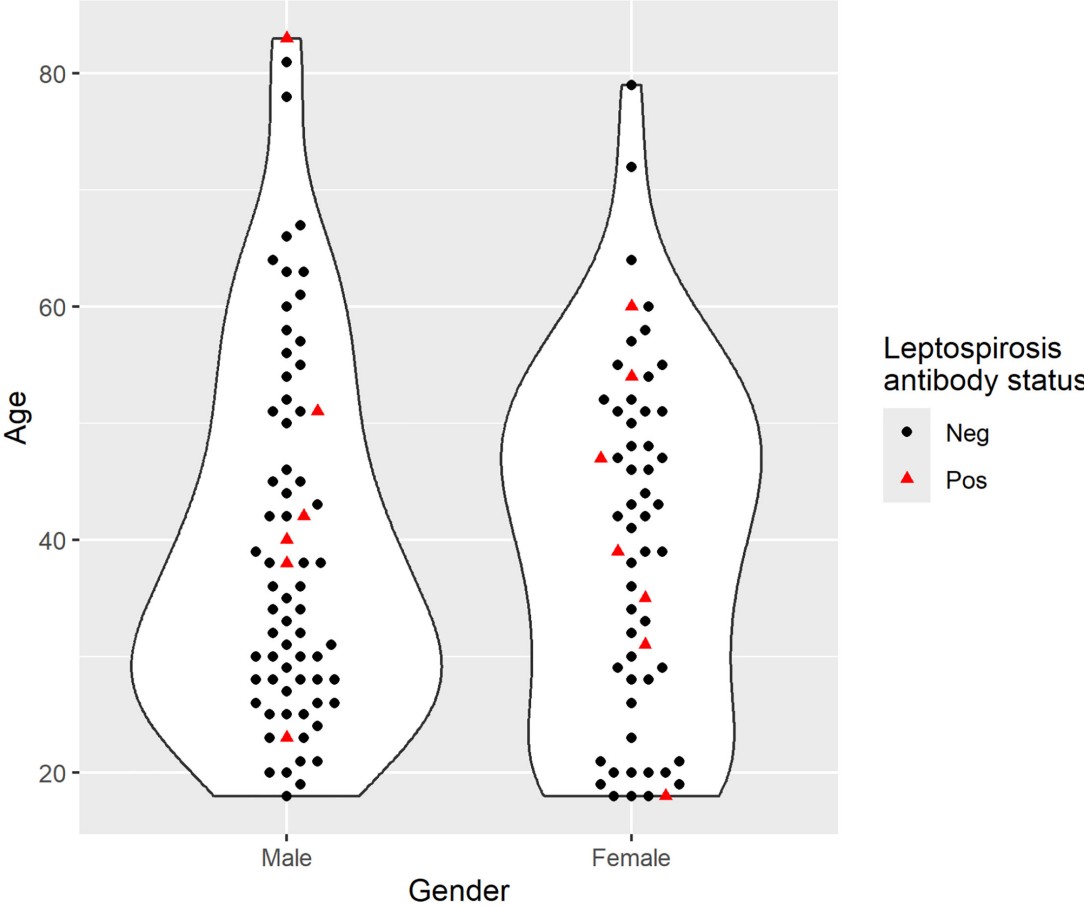

**Appendix 1—figure 2.** Distribution of *Leptospirosis* antibody status (serological status) by gender and age.

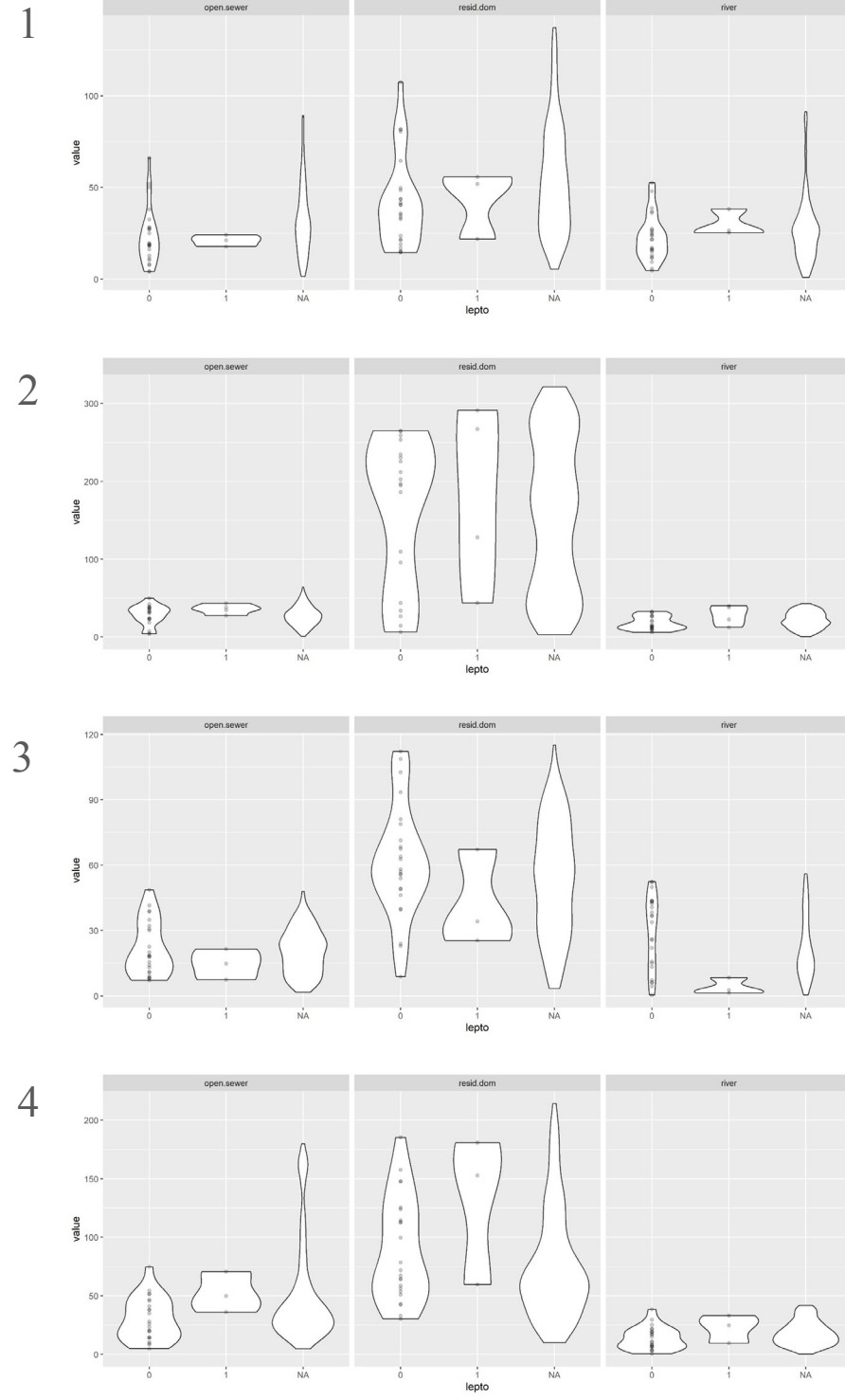

**Appendix 1—figure 3.** Distribution of nearest distance to each of the environmental factors being analysed (central stream, open sewer points, and domestic rubbish piles) by serological status (*x*-axis) and study area (1: NVS, 2: ARE, 3: JSI, 4: CAL). NA represents the rest of households in the study area that did not take part in movement analysis.

