## [Editor Report · eLife Assessment]

This study makes a novel and **valuable** contribution by adapting step selection functions, traditionally used in animal ecology, to explore human movement and environmental risk exposure in urban slums, offering a promising framework for spatial epidemiology, particularly regarding leptospirosis. The integration of GPS telemetry with environmental data and the stratification by gender and serostatus are notable strengths that enhance the study's relevance for public health applications. The strength of evidence is **compelling**.

---

## [Referee Report · Reviewer #1 (Public review)]

Summary:

The study investigated how individuals living in urban slums in Salvador, Brazil, interact with environmental risk factors, particularly focusing on domestic rubbish piles, open sewers, and a central stream. The study makes use of the step selection functions using telemetry data, which is a method to estimate how likely individuals move towards these environmental features, differentiating among groups by gender, age, and leptospirosis serostatus. The results indicated that women tended to stay closer to the central stream while avoiding open sewers more than men. Furthermore, individuals who tested positive for leptospirosis tended to avoid open sewers, suggesting that behavioral patterns might influence exposure to risk factors for leptospirosis, hence ensuring more targeted interventions.

Strengths:

(1) The use of step selection functions to analyze human movement represents an innovative adaptation of a method typically used in animal ecology. This provides a robust quantitative framework for evaluating how people interact with environmental risk factors linked to infectious diseases (in this case, leptospirosis).

(2) Detailed differentiation by gender and serological status allows for nuanced insights, which can help tailor targeted interventions and potentially improve public health measures in urban slum settings.

(3) The integration of real-world telemetry data with epidemiological risk factors supports the development of predictive models that can be applied in future infectious disease research, helping to bridge the gap between environmental exposure and health outcomes.

---

## [Referee Report · Reviewer #2 (Public review)]

Summary:

Pablo Ruiz Cuenca et al. conducted a GPS logger study with 124 adult participants across four different slum areas in Salvador, Brazil, recording GPS locations every 35 seconds for 48 hours. The aim of their study was to investigate step-selection models, a technique widely used in movement ecology to quantify contact with environmental risk factors for exposure to leptospires (open sewers, community streams, and rubbish piles). The authors built two different types of models based on distance and based on buffer areas to model human environmental exposure to risk factors. They show differences in movement/contact with these risk factors based on gender and seropositivity status. This study shows the existence of modest differences in contact with environmental risk factors for leptospirosis at small spatial scales based on socio-demographics and infection status.

Strengths:

The authors assembled a rich dataset by collecting human GPS logger data, combined with field-recorded locations of open sewers, community streams, and rubbish piles, and testing individuals for leptospirosis via serology. This study was able to capture fine-scale exposure dynamics within an urban environment and shows differences by gender and seropositive status, using a method novel to epidemiology (step selection).

[Editors' note: I have reviewed the authors' revised submission and confirm that they have adequately addressed the reviewers' comments for this manuscript.]

---

## [Author Response]

The following is the authors’ response to the original reviews.

**Reviewer #1 (Public review):**
Summary:The study investigated how individuals living in urban slums in Salvador, Brazil, interact with environmental risk factors, particularly focusing on domestic rubbish piles, open sewers, and a central stream. The study makes use of the step selection functions using telemetry data, which is a method to estimate how likely individuals move towards these environmental features, differentiating among groups by gender, age, and leptospirosis serostatus. The results indicated that women tended to stay closer to the central stream while avoiding open sewers more than men. Furthermore, individuals who tested positive for leptospirosis tended to avoid open sewers, suggesting that behavioral patterns might influence exposure to risk factors for leptospirosis, hence ensuring more targeted interventions.Strengths:(1) The use of step selection functions to analyze human movement represents an innovative adaptation of a method typically used in animal ecology. This provides a robust quantitative framework for evaluating how people interact with environmental risk factors linked to infectious diseases (in this case, leptospirosis).(2) Detailed differentiation by gender and serological status allows for nuanced insights, which can help tailor targeted interventions and potentially improve public health measures in urban slum settings.(3) The integration of real-world telemetry data with epidemiological risk factors supports the development of predictive models that can be applied in future infectious disease research, helping to bridge the gap between environmental exposure and health outcomes.Weaknesses:(1) The sample size for the study was not calculated, although it was a nested cohort study.

We thank Reviewer #1 for highlighting this weakness. We will make sure that this is explained in the next version of the manuscript. At the time of recruiting participants, we found no literature on how to perform a sample size calculation for movement studies involving GPS loggers and associated methods of analysis. Therefore, we aimed to recruit as many individuals as possible within the resource constraints of the study.

“Participants who were already enrolled in the cohort study were recruited to take part in the movement analysis study. At the time of recruitment, we found no published scientific studies detailing how to perform sample size calculations for research using GPS data in humans. Therefore, we opted to use convenience sampling instead. A target of 30 people per study area, balanced by gender and blind to their serological status, was chosen for this study.” [Lines 163 - 169]

(2) The step‐selection functions, though a novel method, may face challenges in fully capturing the complexity of human decision-making influenced by socio-cultural and economic factors that were not captured in the study.

We agree with Reviewer #1 that this model may fail to capture the full breadth of human decisionmaking when it comes to moving through local environments. We included a section discussing the aspect of violence and how this influences residents’ choices, along with some possibilities on how to record and account for this. Although it is outside of the scope of this study, we believe that coupling these quantitative methods with qualitative studies would provide a comprehensive understanding of movement in these areas.

(3) The study's context is limited to a specific urban slum in Salvador, Brazil, which may reduce the generalizability of its findings to other geographical areas or populations that experience different environmental or socio-economic conditions.

We thank the reviewer for highlighting this limitation. We have made this more clear in the discussion section:

“As a result, the findings are biased towards the more represented individuals, limiting their generalisability. Additionally, all participants are from specific areas in Salvador, which may further limit the generalisability to similar contexts.” [Lines 561 - 564]

(4) The reliance on self-reported or telemetry-based movement data might include some inaccuracies or biases that could affect the precision of the selection coefficients obtained, potentially limiting the study's predictive power.

We agree that telemetry data has inherent inaccuracies, which we have tried to account for by using only those data points within the study areas. We would like to clarify that there is no self-reported movement data used in this study. All movement data was collected using GPS loggers.

(5) Some participants with less than 50 relocations within the study area were excluded without clear justification, see line 149.

We found that the SSF models would not run properly if there weren’t enough relocations. Therefore, we decided to remove these individuals from the analysis. They are also removed from any descriptive statistics presented. We have now clarified this in the manuscript.

“Individuals with less than 50 relocations within the study area were excluded from the analysis to ensure good model convergence. Details of these excluded individuals can be found in Supplementary Material I.” [Lines 183 – 186]

(6) Some figures are not clear (see Figure 4 A & B).

We have improved the resolution of the image and believe it is more clear now. Please let us know if the resolution still is not clear enough.

(7) No statement on conflict of interest was included, considering sponsorship of the study.

The conflict of interest forms for each author were sent to eLife separately. I believe these should be made available upon publication, but please reach out if these need to be re-sent.

**Reviewer #2 (Public review):**
Summary:Pablo Ruiz Cuenca et al. conducted a GPS logger study with 124 adult participants across four different slum areas in Salvador, Brazil, recording GPS locations every 35 seconds for 48 hours. The aim of their study was to investigate step-selection models, a technique widely used in movement ecology to quantify contact with environmental risk factors for exposure to leptospires (open sewers, community streams, and rubbish piles). The authors built two different types of models based on distance and based on buffer areas to model human environmental exposure to risk factors. They show differences in movement/contact with these risk factors based on gender and seropositivity status. This study shows the existence of modest differences in contact with environmental risk factors for leptospirosis at small spatial scales based on socio-demographics and infection status.Strengths:The authors assembled a rich dataset by collecting human GPS logger data, combined with fieldrecorded locations of open sewers, community streams, and rubbish piles, and testing individuals for leptospirosis via serology. This study was able to capture fine-scale exposure dynamics within an urban environment and shows differences by gender and seropositive status, using a method novel to epidemiology (step selection).Weaknesses:Due to environmental data being limited to the study area, exposure elsewhere could not be captured, despite previous research by Owers et al. showing that the extent of movement was associated with infection risk. Limitations of step selection for use in studying human participants in an urban environment would need to be explicitly discussed.

The environmental factors used in the study required research teams to visit the sites and map the locations. Given that individuals travelled throughout the city of Salvador, performing this task at a large scale would be unachievable. Therefore, we limited the data to only those points within the study area boundaries to avoid any biases from interactions with unrecorded environmental factors.

**Reviewing Editor Comments:**
The manuscript would benefit from clearer articulation of SSF assumptions, data exclusions, and buffer choices, as well as improvements in figure clarity, to strengthen its generalizability and impact.

Please see replies to Reviewer #2 below regarding the assumptions (2.3), data exclusions (2.1) and buffer choices (2.2). We have improved Figure 4 clarity, please let us know if this is not sufficient.

**Reviewer #1 (Recommendations for the authors):**
(1) Provide comprehensive details on telemetry data collection for improved data quality and reproducibility.

Details for this are included under the “Methods/GPS Data” section. We have included a sentence to explain that we used to GPS device manufacturer’s software to programme them. We believe this provides enough information on how to collect the data for reproducibility, but please let us know if there is further information that we could provide.

“Individuals who consented to take part in this study were asked to wear GPS loggers for continuous periods of up to 48 hours, which could be repeated. The GPS loggers used were i-got U GT-600, set to record their location every 35 seconds. We used the manufacturer’s software to programme the devices. Data were collected between March and November 2022.” [Lines 172 - 176]

(2) Check all figures and improve on clarity (see Figure 4).

We have updated Figure 4 and believe the resolution is better now. Please let us know if this it not the case from the readers perspective.

(3) Revisit sentence structures to improve readability and reduce overly complex phrasing.

We have reviewed the manuscript and made some changes to improve readability.

**Reviewer #2 (Recommendations for the authors):**
I thank Ruiz Cuenca et al. for putting together this interesting manuscript on the use of step selection functions for understanding exposure to leptospires in urban Brazil. I thoroughly enjoyed reading it and have a few suggestions that may improve the manuscript.I also apologise, but I was not able to find some of the supplementary materials, for instance, Supplementary Material I. That may have been my oversight.

To eLife: These should have been included with the submitted manuscript file. Please let me know if it has to be resubmitted to eLife.

(1) Descriptive statisticsSome more descriptive statistics would be helpful. For instance, what was the leptospirosis infection status of the six individuals who were removed due to having <50 points inside the area? As part of the analysis relies on exposure, defined as GPS locations within a 20m buffer of open sewers, community streams, and rubbish piles, it would be good to have some descriptive statistics around this. How many visits to these different sites did people make, and how did these statistics vary by study area, age, gender, and leptospirosis infection status?

We thank Reviewer #2 for highlighting this. Thanks to their comment, we noticed a mistake in the code which excluded more individuals from the summary statistics table than were actually excluded from the full analysis. There were only 2 individuals that had less than 50 relocations across the whole day (5 am to 9 pm) which were excluded from further analysis. The mistake has been rectified and the summary statistics updated. (see table 1)

We have included the demographic details of excluded participants as a table in the supplementary material, which we have referenced to in the manuscript. We have also explained that the exclusion is to aid model convergence, as we found that too few relocations would result in SSF models not working properly.

“Individuals with less than 50 relocations within the study area were excluded from the analysis to ensure good model convergence. Details of these excluded individuals can be found in Supplementary Material I.” [Lines 183 – 186]

We have also now included a table (Table 2), to show more descriptive statistics of how much time individuals spent within each of the environmental buffers.

(2) Definitions of buffersI was surprised that the authors chose a 20m buffer for each factor but 10m around the household.Could this be more clearly justified, especially given that there will be location errors in both the GPS location point and the GPS logger points? These buffers do appear quite small, particularly in an urban environment where obstruction from buildings can be expected to yield substantial GPS errors.

The 20 meter buffer represents an intense interaction with the point of interest. This distance was decided after visiting the sites and seeing the points of interest in person. The 10 meter buffer accounts for the size of dwellings in these areas. We have included these explanations in the new manuscript:

“The buffer rasters, one for each factor, were created using a 20 meter buffer around each reference point. The size of this buffer was decided after visiting the study areas and represented an area within which it could be considered a strong interaction with the point of interest.” [Lines 198 – 202]

“Buffer rasters were also created for each individual’s household location, with a 10 meter buffer around each location.This represented space within and immediately outside each house. This buffer size accounted for the size of dwellings in these study areas.” [Lines 205 - 208]

(3) Assumptions of the step selection functionStep selection functions (SSFs) rely on a number of assumptions. Whether these assumptions are met needs to be critically discussed within the article. (For a discussion of the assumptions, I am relying on points raised in this article: Integrated step selection analysis: bridging the gap between resource selection and animal movement (2015): Tal Avgar, Jonathan R. Potts, Mark A. Lewis, Mark S. Boyce, DOI: https://doi.org/10.1111/2041-210X.12528).First, SSFs typically assume each step is independent, conditional only on the previous step (Markovian process). This is violated in circular movements, for instance. Circular movements are highly likely in human movement as people will leave and return to their homes during the day. While this is partially addressed by conducting separate analyses by time of day, circular journeys can still exist within these segments.Second, SSFs do not account for goal-oriented behaviour like intentional destination-seeking. So, for instance, when someone executes a plan to visit a specific stream to fetch drinking water, such behaviour is poorly approximated using SSFs because SSFs compare observed steps to random alternatives drawn from a movement kernel, assuming movement is opportunistic rather than intentional.

This is true of SSF that do not include movement attributes. However, in our SSF we have included both step lengths and turning angles, which, according to Avgar et al, should be enough to account for this goal-oriented behaviour. It may be clearer to call the model an integrated step selection function (iSSF), as they do in Avgar et al., which we can change in the next version of the manuscript.

Third, turning angles in human movement are often sharp due to regular street layout, which can violate the assumptions of SSFs, which usually assume smooth, correlated movement.As this paper proposes SSFs as a novel method to measure exposure to environmentally transmitted pathogens, a discussion on the extent to which assumptions of SSFs are valid for this purpose should be included in the paper.

We thank Reviewer #2 for highlighting these points. We have included a section discussing these assumptions in detail:

“Additionally, these models have some underlying assumptions that may be violated in this study. Step-selection functions assume each step is independent, conditioned on the previous step. This can be violated by circular journeys. Although we attempted to account for these by analysing specific periods of the day, a higher temporal resolution of analysis may be needed if circular journeys are still present within each period. Another assumption is that movement is smooth through the environment. In urban environments this may not hold true, as street layouts may force sharp corners in movements. The effect of violating this assumption is not immediately clear and requires further methodological research to understand its significance. Finally, we assumed that by including movement characteristics (step lengths and turning angles) into our models, we were accounting for goal-oriented behaviour. These assumptions need to be considered in future studies that attempt to use step-selection functions to analyse human mobility.” [Lines 593 - 607]

(4) AbstractWhile it is highlighted in the abstract that this "study introduces a novel method for analysing human telemetry data in infectious disease research, providing critical insights for targeted interventions", I did not see any discussion about how the findings can inform interventions.

We thank Reviewer #2 for highlighting this. We have now removed this wording from the abstract to avoid misunderstanding.

(5) Effect sizesIt would have helped me if there had been some discussion around the size of these effects. Especially for the distance-based models, the effects seem very small. Maybe this is a misinterpretation on my part, but it would help to contextualise if the observed effect were small or large.

We agree with Reviewer #2 on this point and have now included a paragraph explaining that these effect sizes are indeed very small. We believe that this may be linked to the spatial scale of the rasters used (1 meter), as the selection coefficients represent changes with regards to increasing distances of 1 meter. This may not be that significant for human mobility. However, given the focus on analysing fine scale movement, we decided to keep the spatial scale of the rasters as small as possible.

“It is important to highlight that the effect sizes of the selection coefficients for the distance based rasters are very small and could be considered negligible. This may be linked to the spatial scale used, as these values represent increases of 1 meter. A coarser scale may have produced larger effect sizes that may have been easier to conceptualise. However, given the focus on fine-scale movement, we decided to keep this spatial scale for the analysis.” [Lines 421 - 427]